# ADAPTING BEHAVIOUR FOR LEARNING PROGRESS

## ABSTRACT

Determining what experience to generate to best facilitate learning (i.e. exploration) is one of the distinguishing features and open challenges in reinforcement learning. The advent of distributed agents that interact with parallel instances of the environment has enabled larger scale and greater flexibility, but has not removed the need to tune or tailor exploration to the task, because the ideal data for the learning algorithm necessarily depends on its process of learning. We propose to dynamically adapt the data generation by using a non-stationary multi-armed bandit to optimize a proxy of the learning progress. The data distribution is controlled via modulating multiple parameters of the policy (such as stochasticity, consistency or optimism) without significant overhead. The adaptation speed of the bandit can be increased by exploiting the factored modulation structure. We demonstrate on a suite of Atari 2600 games how this unified approach produces results comparable to per-task tuning at a fraction of the cost.

## 1 INTRODUCTION

Reinforcement learning (RL) is a general formalism modelling sequential decision making. It aspires to be broadly applicable, making minimal assumptions about the task at hand and reducing the need for prior knowledge. By learning behaviour from scratch, it has the potential to surpass human expertise or tackle complex domains where human intuition is not applicable. In practice, however, generality is often traded for performance and efficiency, with RL practitioners tuning algorithms, architectures and hyper-parameters to the task at hand (Hessel et al., 2019). A side-effect of this is that the resulting methods can be brittle, or difficult to reliably reproduce (Nagarajan et al., 2018).

Exploration is one of the main aspects commonly designed or tuned specifically for the task being solved. Previous work has shown that large sample-efficiency gains are possible, for example, when the exploratory behaviour's level of stochasticity is adjusted to the environment's hazard rate (García & Fernández, 2015), or when an appropriate prior is used in large action spaces (Dulac-Arnold et al., 2015; Czarnecki et al., 2018; Vinyals et al., 2019). Ideal exploration in the presence of function approximation should be agent-centred. It ought to focus more on generating data that supports the learning of agent at its current parameters $\theta$, rather than making progress on objective measurements of information gathering. A useful notion here is *learning progress* ($LP$), defined as the improvement of the learned policy $\pi_\theta$ (Section 3).

The agent's source of data is its behaviour policy. Beyond the conventional RL setting of a single stream of experience, distributed agents that interact with parallel copies of the environment can have multiple such data sources (Horgan et al., 2018). In this paper, we restrict ourselves to the setting where all behaviour policies are derived from a single set of learned parameters $\theta$, for example when $\theta$ parameterises an action-value function $Q_\theta$. Consequently the behaviour policies are given by $\pi(Q_\theta, z)$, where each *modulation* $z$ leads to meaningfully different behaviour. This can be guaranteed if $z$ is semantic (e.g. degree of stochasticity) and consistent across multiple time-steps. The latter is achieved by holding $z$ fixed throughout each episode (Section 2).

We propose to estimate a proxy that is indicative of future learning progress, $f(z)$ (Section 3), separately for each modulation $z$, and to *adapt* the distribution over modulations to maximize $f$, using a non-stationary multi-armed bandit that can exploit the factored structure of the modulations (Section 4). Figure 1 shows a diagram of all these components. This results in an autonomous adaptation of behaviour to the agent's stage of learning (Section 5), varying across tasks and across time, and reducing the need for hyper-parameter tuning.

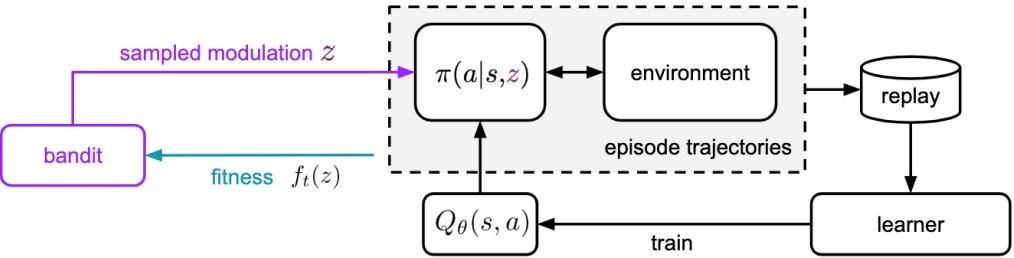

Figure 1: The overall architecture is based on a distributed deep RL agent (black). In addition, modulations $z$ are sampled once per episode and modulate the behaviour policy. A bandit (purple) adapts the distribution over $z$ based on a fitness $f_t(z)$ that approximates learning progress (cyan).

## 2 MODULATED BEHAVIOUR

As usual in RL, the objective of the agent is to find a *policy* $\pi$ that maximises the $\gamma$-discounted expected *return* $G_t \doteq \sum_{i=0}^{\infty} \gamma^i R_{t+i}$, where $R_t$ is the reward obtained during the transition from time $t$ to $t+1$. A common way to address this problem is to use methods that compute the *action-value function* $Q^\pi$ given by $Q^\pi(s,a) \doteq \mathbb{E}[G_t|s,a]$, i.e. the expected return when starting from state $s$ with action $a$ and then following $\pi$ (Puterman, 1994).

A richer representation of $Q^\pi$ that aims to capture more information about the underlying distribution of $G_t$ has been proposed by Bellemare et al. (2017), and extended by Dabney et al. (2018). Instead of approximating only the mean of the return distribution, we approximate a discrete set of $n$ quantile values $q_\nu$ (where $\nu \in \{\frac{1}{2n}, \frac{3}{2n}, \dots, \frac{2n-1}{2n}\}$) such that $\mathbb{P}(Q^\pi \leq q_\nu) = \nu$. Outside the benefits in performance and representation learning (Such et al., 2018), these quantile estimates provide a way of inducing risk-sensitive behaviour. We approximate all $q_\nu$ using a single deep neural network with parameters $\theta$, and define the evaluation policy as the greedy one with respect to the mean estimate:

$$\pi_\theta(\cdot|s) \in \arg\max_a \frac{1}{n} \sum_\nu q_\nu(s,a).$$

The behaviour policy is the central element of exploration: it generates exploratory behaviour (and experience therefrom) which is used to learn $\pi_\theta$; ideally in such a way as to reduce the total amount of experience required to achieve good performance. Instead of a single monolithic behaviour policy, we propose to use a *modulated* policy to support parameterized variation. Its modulations $z$ should satisfy the following criteria: they need to (i) be impactful, having a direct and meaningful effect on generated behaviour; (ii) have small dimensionality, as to quickly adapt to the needs of the learning algorithm, and interpretable semantics to ease the choice of viable ranges and initialisation; and (iii) be frugal, in the sense that they are relatively simple and computationally inexpensive to apply. In this work, we consider five concrete types of such modulations:

**Temperature:** a Boltzmann softmax policy based on action-logits, modulated by temperature, $T$.

**Flat stochasticity:** with probability $\epsilon$ the agent ignores the action distribution produced by the softmax, and samples an action uniformly at random ($\epsilon$-greedy).

**Per-action biases:** action-logit offsets, $\boldsymbol{b}$, to bias the agent to prefer some actions.

**Action-repeat probability:** with probability $\rho$, the previous action is repeated (Machado et al., 2017). This produces chains of repeated actions with expected length $\frac{1}{1-\rho}$.

**Optimism:** as the value function is represented by quantiles $q_\nu$, the aggregate estimate $Q_\omega$ can be parameterised by an optimism exponent $\omega$, such that $\omega = 0$ recovers the default flat average, while positive values of $\omega$ imply optimism and negative ones pessimism. When near risk-neutral, our simple risk measure produces qualitatively similar transforms to those of Wang (2000).

We combine the above modulations to produce the overall $z$-modulated policy

$$\pi(a|s,z) \doteq (1-\epsilon)(1-\rho)\frac{e^{\frac{1}{T}(Q_\omega(s,a)+\boldsymbol{b}_a)}}{\sum_{a'\in\mathcal{A}} e^{\frac{1}{T}(Q_\omega(s,a')+\boldsymbol{b}_{a'})}} + \frac{\epsilon(1-\rho)}{|\mathcal{A}|} + \rho\mathbb{I}_{a=a_{t-1}},$$

where $z \doteq (T, \epsilon, \boldsymbol{b}, \rho, \omega)$, $\mathbb{I}_x$ is the indicator function, and the optimism-aggregated value is

$$Q_\omega \doteq \frac{\sum_\nu e^{-\omega\nu} q_\nu}{\sum_\nu e^{-\omega\nu}}.$$

Now that the behaviour policy can be modulated, the following two sections discuss the criteria and mechanisms for choosing modulations $z$.

## 3 EXPLORATION & THE EFFECTIVE ACQUISITION OF INFORMATION

A key component of a successful reinforcement learning algorithm is the ability to acquire experience (information) that allows it to make expeditious progress towards its objective of learning to act in the environment in such a way as to optimise returns over the relevant (potentially discounted) horizon. The types of experience that most benefit an agent's ultimate performance may differ qualitatively throughout the course of learning — a behaviour modulation that is beneficial in the beginning of training often enough does not carry over to the end, as illustrated by the analysis in Figure 5. However, this analysis was conducted in hindsight, and in general how to generate such experience optimally — optimal exploration in any environment — remains an open problem.

One approach is to require exploration to be in service of the agent's future learning progress ($LP$), and to optimise this quantity during learning. Although there are multiple ways of defining learning progress, in this work we opted for a task-related measure, namely the improvement of the policy in terms of expected return. This choice of measure corresponds to the local steepness of the learning curve of the evaluation policy $\pi_\theta$,

$$LP_t(\Delta\theta) \doteq \mathbb{E}_{s_0} \left[ V^{\pi_{\theta_t + \Delta\theta}}(s_0) - V^{\pi_{\theta_t}}(s_0) \right], \tag{1}$$

where the expectation is over start states $s_0$, the value $V^\pi(s) = \mathbb{E}_\pi[\sum \gamma^i R_i | s_0 = s]$ is the $\gamma$-discounted return one would expected to obtain, starting in state $s$ and following policy $\pi$ afterwards, and $\Delta\theta$ is the change in the agent's parameters. Note that this is still a limited criterion, as it is myopic and might be prone to local optima.

As prefaced in the last section, our goal here is to define a mechanism that can switch between different behaviour modulations depending on which of them seems most promising at this point in the training process. Thus in order to adapt the distribution over modulations $z$, we want to assess the expected LP when learning from data generated according to $z$-modulated behaviour:

$$LP_t(z) \doteq \mathbb{E}_{\tau \sim \pi_{\theta_t}(z)} [LP_t(\Delta\theta(\tau, t))],$$

with $\Delta\theta(\tau, t)$ the weight-change of learning from trajectory $\tau$ at time $t$. This is a subjective utility measure, quantifying how useful $\tau$ is for a particular learning algorithm, at this stage in training.

**Proxies for learning progress:** While $LP(z)$ is a simple and clear progress metric, it is not readily available during training, so that in practice, a *proxy fitness* $f_t(z) \approx LP_t(z)$ needs to be used. A key practical challenge is to construct $f_t$ from inexpensively measurable proxies, in a way that is sufficiently informative to effectively adapt the distribution over $z$, while being robust to noise, approximation error, state distribution shift and mismatch between the proxies and learning progress. The ideal choice of $f(z)$ is a matter of empirical study, and this paper only scratches the surface on this topic.

After some initial experimentation, we opted for the simple proxy of empirical (undiscounted) episodic return: $f_t(z) = \sum_{a_i \sim \pi(Q_{\theta_t}, z)} R_i$. This is trivial to estimate, but it departs from $LP(z)$ in a number of ways. First, it does not contain learner-subjective information, but this is partly mitigated through the joint use of with prioritized replay (see Section 5.1) that over-samples high error experience. Another potential mechanism by which the episodic return can be indicative of *future* learning is because an improved policy tends to be *preceded* by some higher-return episodes – in general, there is a lag between best-seen performance and reliably reproducing it. Second, the fitness is based on absolute returns not differences in returns as suggested by Equation 1; this makes no difference to the relative orderings of $z$ (and the resulting probabilities induced by the bandit), but it has the benefit that the non-stationarity takes a different form: a difference-based metric will

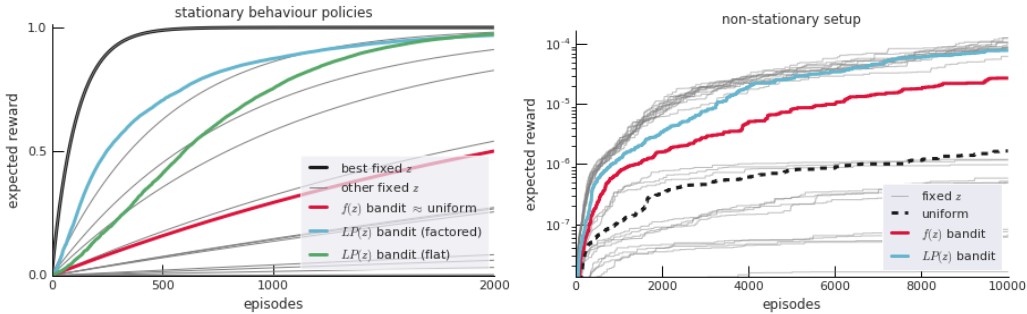

Figure 2: LavaWorld experiments, with 31 distinct modulations $z$. **Left**: If the candidate behaviours are stationary (not affected by learning), the bandit nearly recovers the performance of the best fixed choice. The factored variant (blue) is more effective than a flat discretization (green). Using the true $LP(z)$ instead of a less informative proxy (red) has a distinct effect: in the stationary sparse-reward case this makes the bandit equivalent to uniform sampling. **Right**: in a non-stationary setting where the modulated Q-values are learned over time, the dynamics are more complex, and even the bandit with ideal fitness again comes close to the best fixed choice, while uniform or proxy-based variants struggle more (see Appendix B).

appear stationary if the policy performance keeps increasing at a steady rate, but such a policy must be changing significantly to achieve that progress, and therefore the selection mechanism should keep revisiting other modulations. In contrast, our absolute fitness naturally has this effect when paired with a non-stationary bandit, as described in the next section.

## 4 NON-STATIONARY BANDIT TAILORED TO LEARNING PROGRESS

The most effective modulation scheme may differ throughout the course of learning. Instead of applying a single fixed modulation or fixed blend, we propose an *adaptive* scheme, in which the choice of modulation is dynamically based on learning progress. The adaptation process is based on a *non-stationary* multi-armed bandit (Besbes et al., 2014; Raj & Kalyani, 2017), where each arm corresponds to a behaviour modulation $z$. The non-stationarity reflects the nature of the learning progress $LP_t(z)$ which depends on the time $t$ in training through the parameters $\theta_t$.

Because of non-stationarity, the core challenge for such bandit is to identify good modulation arms quickly, while only having access to a noisy, indirect proxy $f_t(z)$ of the quantity of interest $LP_t(z)$. However, our setting also presents an unusual advantage: the bandit does not need to identify the *best* $z$, as in practice it suffices to spread probability among all arms that produce reasonably useful experience for learning.

Concretely, our bandit samples a modulation $z \in \{z_1, \dots, z_K\}$ according to the probability that it results in higher than usual fitness (measured as the mean over a recent length-$h$ window):

$$\mathbb{P}_t(z) \propto \mathbb{P}(f_t(z) \geq m_t), \quad \text{where} \quad m_t \doteq \frac{1}{h} \sum_{t'=t-h}^{t-1} f_{t'}(z_{t'}).$$

Note that $m_t$ depends on the payoffs of the actually sampled modulations $z_{t-h:t-1}$, allowing the bandit to become progressively more selective (if $m_t$ keeps increasing).

**Estimation:** For simplicity, $\mathbb{P}_t(z)$ is inferred based on the empirical data within a recent time window of the same horizon $h$ that is used to compute $m_t$. Concretely, $\mathbb{P}_t(z) \doteq \mu_t(z)/\sum_{z'} \mu_t(z')$ with the preferences $\mu_t(z) \approx \mathbb{P}(f_t(z) \geq m_t)$ defined as

$$\mu_t(z) \doteq \frac{\frac{1}{2} + \sum_{t'=t-h}^{t-1} \mathbb{I}_{f_{t'}(z_t') \geq m_t} \mathbb{I}_{z_{t'}=z}}{1 + n(z, h)}$$

where $n(z, h)$ is the number of times that $z$ was chosen in the corresponding time window. We encode a prior preference of $\frac{1}{2}$ in the absence of other evidence, as an additional (fictitious) sample.

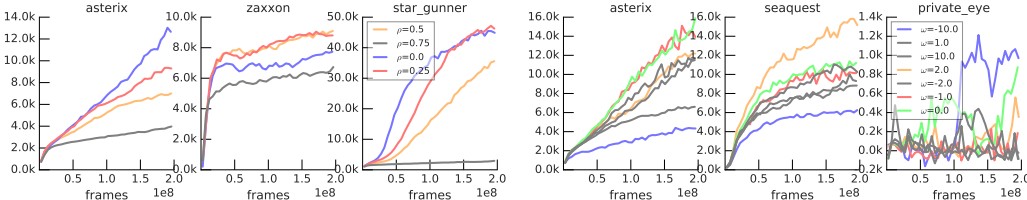

Figure 3: The best fixed choices of a modulation $z$ vary per game, and this result holds for multiple classes of modulations. **Left**: Repeat probabilities $\rho$. **Right:** Optimism $\omega$.

**Adaptive horizon:** The choice of $h$ can be tuned as a hyper-parameter, but in order to remove all hyper-parameters from the bandit, we adapt it online instead. The update is based on a regression accuracy criterion, weighted by how often the arm is pulled. For the full description, see Appendix A.

**Factored structure:** As we have seen in Section 2, our concrete modulations $z$ have additional factored structure that can be exploited. For that we propose to use a separate *sub-bandit* (each defined as above) for each dimension $j$ of $z$. The full modulation $z$ is assembled from the $z_j$ independently sampled from the sub-bandits. This way, denoting by $K_j$ the number of arms for $z_j$, the total number of arms to model is $\sum_j K_j$, which is a significant reduction from the number of arms in the single flattened space $\prod_j K_j$. This allows for dramatically faster adaptation in the bandit (see Figure 2). On the other hand, from the perspective of each sub-bandit, there is now another source of non-stationarity due to other sub-bandits shifting their distributions.

## 5 EXPERIMENTS

The central claim of this paper is that the best fixed hyper-parameters in hindsight for behaviour differ widely across tasks, and that an adaptive approach obtains similar performance to the best choice without costly per-task tuning. We report a broad collection of empirical results on Atari 2600 (Bellemare et al., 2013) that substantiate this claim, and validate the effectiveness of the proposed components. From our results, we distill qualitative descriptions of the adaptation dynamics. To isolate effects, independent experiments may use all or subsets of the dimensions of $z$.

Two initial experiments in a toy grid-world setting are reported in Figure 2. They demonstrate that the proposed bandit works well in both stationary and non-stationary settings. Moreover, they highlight the benefits of using the exact learning progress $LP(z)$, and the gap incurred when using less informative proxies $f(z)$. They also indicate that the factored approach can deliver a substantial speed-up. Details of this setting are described in Appendix B.

### 5.1 EXPERIMENTAL SETUP: ATARI

Our Atari agent is a distributed system inspired by Impala (Espeholt et al., 2018) and Ape-X (Horgan et al., 2018), consisting of one learner (on GPU), multiple actors (on CPUs), and a bandit providing modulations to the actors. On each episode $t$, an actor queries the bandit for a modulation $z_t$, and the learner for the latest network weights $\theta_t$. At episode end, it reports a fitness value $f_t(z_t)$ to the bandit, and adds the collected experience to a replay table for the learner. For stability and reliability, we enforce a fixed ratio between experience generated and learning steps, making actors and learner run at the same pace. Our agents learn a policy from 200 million environment frames in 10-12h wall-clock time (compared to a GPU-week for the state-of-art Rainbow agent (Hessel et al., 2018)).

Besides distributed experience collection (i.e., improved experimental turnaround time), algorithmic elements of the learner are similar to Rainbow: the updates use multi-step double Q-learning, with distributional quantile regression (Dabney et al., 2018) and prioritized experience replay (Schaul et al., 2015). All hyper-parameters (besides those determined by $z$) are kept fixed across all games and all experiments; these are listed in Appendix C alongside default values of $z$. These allow us to generate competitive baseline results ($118 \pm 6\%$ median human-normalised score) with a so-called *reference* setting (solid black in all learning curves) which sets the exploration parameters to that is most commonly used in the literature ($\epsilon = 0.01$, $\omega = 0$, $T = 0$, $\mathbf{b} = 0$, $\rho = 0$).

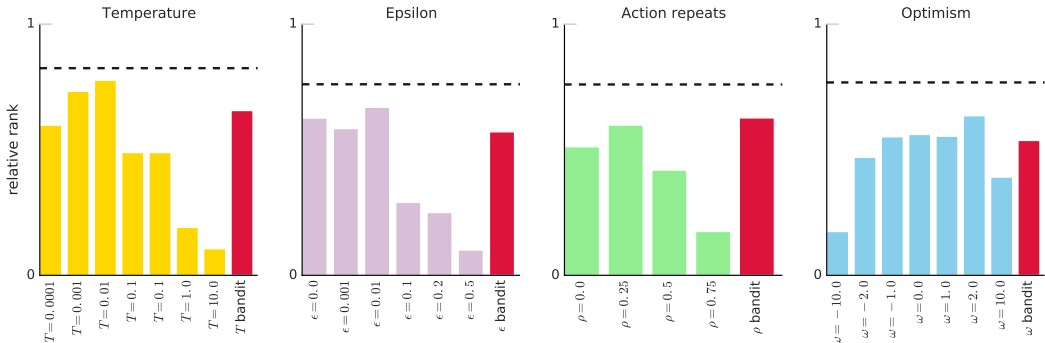

Figure 4: No fixed arm always wins. Shown are normalized *relative ranks* (see text) of different fixed arms (and curated bandit), where ranking is done separately per modulation class (subplot). The score of 1 would be achieved by an arm that performs best across all seeds (0 if it is always the worst). Dashed lines are ranks of an oracle that could pick the best fixed arm in hindsight *per game*. The gap between the dashed line and the maximum 1 indicates the effect of inter-seed variability, and the gap between the dashed line and the highest bar indicates that different games require different settings. In all four modulation classes, the bandit performance is comparable to one of the best fixed choices.

If not mentioned otherwise, all aggregate results are across 15 games listed in Appendix D and at least $N = 5$ independent runs (seeds). Learning curves shown are evaluations of the greedy policy after the agent has experienced the corresponding number of environment frames. To aggregate scores across these fifteen games we use *relative rank*, an ordinal statistic that weighs each game equally (despite different score scales) and highlights relative differences between variants. Concretely, the performance outcome $G(game, seed, variant)$ is defined as the average return of the greedy policy across the last 10% of the run (20 million frames). All outcomes $G(game, \cdot, \cdot)$ are then jointly ranked, and the corresponding ranks are averaged across seeds. The averaged ranks are normalized to fall between 0 and 1, such that a normalized rank of 1 corresponds to all $N$ seeds of a variant being ranked at the top $N$ positions in the joint ranking. Finally, the relative ranks for each variant are averaged across all games. See also Appendix D.

## 5.2 QUANTIFYING THE TUNING CHALLENGES

It is widely appreciated that the best hyper-parameters differ per Atari game. Figure 3 illustrates this point for multiple classes of modulations (different arms come out on top in different games), while Figure 4 quantifies this phenomenon across 15 games and 4 modulation classes and finds that this effect holds in general.

If early performance were indicative of final performance, the cost of tuning could be reduced. We quantify how much performance would be lost if the best fixed arm were based on the first 10% of the run. Figure 5 shows that the mismatch is often substantial. This also indicates the best choice is *non-stationary*: what is good in early learning may not be good later on — an issue sometimes addressed by hand-crafted schedules (e.g., DQN linearly decreases the value of $\epsilon$ (Mnih et al., 2015)).

Another approach is to choose not to choose, that is, feed experience from the full set of choices to the learner, an approach taken, e.g., in (Horgan et al., 2018). However, this merely shifts the problem, as it in turn necessitates tuning this *set* of choices. Figure 6 shows that the difference between a naive and a carefully *curated* set can indeed be very large (Table 4 in Appendix C lists all these sets).

## 5.3 ADAPTING INSTEAD OF TUNING

It turns out that adapting the distribution over $z$ as learning progresses effectively addresses the three tuning challenges discussed above (per-task differences, early-late mismatch, handling sets). Figure 6 shows that the bandit can quickly suppress the choices of harmful elements in a non-curated set; in other words, the set does not need to be carefully tuned. At the same time, a game-

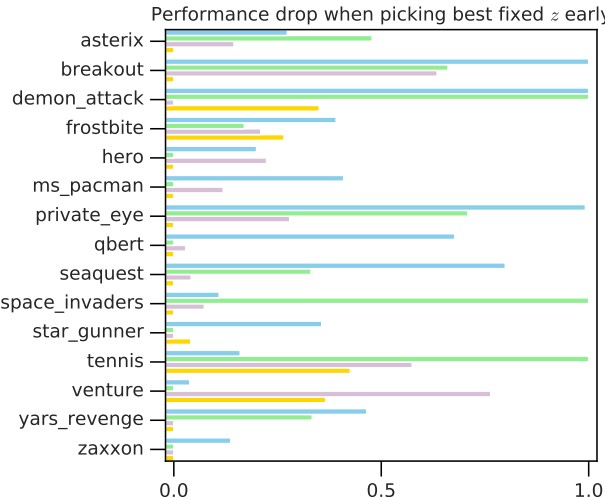

Figure 5: Tuning exploration by early performance is non-trivial. The bar-plot shows the performance drop when choosing a fixed $z$ based on initial performance (first 10% of the run) as compared to the best performance in hindsight (scores are normalized between best and worst final outcome, see Appendix D). This works well for some games but not others, and better for some modulation classes than others (colour-coded as in Figure 6), but overall its not a reliable method.

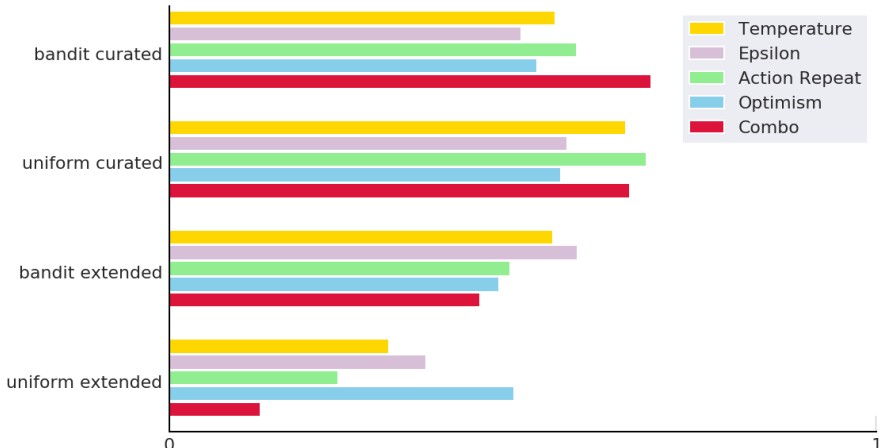

Figure 6: Tuning exploration by choosing a set of modulations is non-trivial. This figure contrasts four settings: the effect of naive ('extended') versus curated $z$-sets, crossed with uniformly sampling or using the bandit, and for each of the four settings results are given for multiple classes of modulations (colour-coded). The length of the bars are normalized relative ranks (higher is better). Note how uniform does well when the set is curated, but performance drops on extended $z$-sets (except for $\omega$-modulations, which are never catastrophic); however, the bandit recovers a similar performance to the curated set by suppressing harmful modulations (the third group is closer to the first than to the fourth). The 'combo' results are for the combinatorial $\epsilon, \rho, \omega$ space, where the effect is even more pronounced.

specific schedule emerges from the non-stationary adaptation, for example recovering an $\epsilon$-schedule reminiscent of the hand-crafted one in DQN (Mnih et al., 2015) (see Figure 17 in Appendix E). Finally, the overall performance of the bandit is similar to that of the best fixed choice, and not far from an "oracle" that picks the best fixed $z$ *per game* in hindsight (Figure 4).

A number of other interesting qualitative dynamics emerge in our setting (Appendix E): action biases are used initially and later suppressed (e.g., on SEAQUEST, Figure 19); the usefulness of action

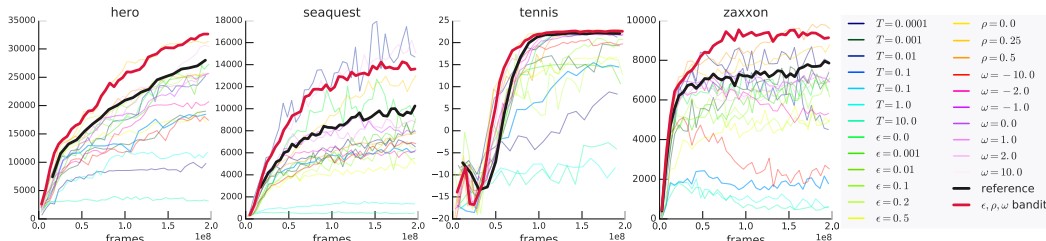

Figure 7: A few games cherry-picked to show that the combinatorial $\epsilon, \rho, \omega$ bandit (thick red) can sometimes outperform all the fixed settings of $z$. The thick black line shows the fixed reference setting $\epsilon = 0.01$.

repeats varies across training (e.g., on H.E.R.O., Figure 18). Figure 16 looks at additional bandit baselines and finds that addressing the non-stationarity is critical (see Appendix E.3).

Finally, our approach generalizes beyond a single class of modulations; all proposed dimensions can adapt simultaneously within a single run, using a factored bandit to handle the combinatorial space. Figure 13 shows this yields similar performance to adapting within one class. In a few games this outperforms the best fixed choice[1] in hindsight; see Figure 6 ('combo') and Figure 7; presumably because of the added dynamic adaptation to the learning process. On the entire set of 57 Atari games, the bandit achieves similar performance ($113 \pm 2\%$ median human-normalized score) to our fixed, tuned reference setting ($118 \pm 6\%$), despite operating on 60 different combinations of modulations.

## 6 RELATED WORK

Here we focus on two facets of our research: its relation to exploration, and hyper-parameter tuning.

First, our work can be seen as building on a rich literature on exploration through intrinsic motivation aimed at maximising learning progress. As the true learning progress is not readily available during training, much of this work targets one of a number of proxies: empirical return (Jaderberg et al., 2017); change in parameters, policy, or value function (Itti & Baldi, 2006); magnitude of training loss (Mirolli & Baldassarre, 2013; Schmidhuber, 1991); error reduction or derivative (Schmidhuber, 1991; Oudeyer et al., 2007); expected accuracy improvement (Misra et al., 2018); compression progress (Schmidhuber, 2008); reduction in uncertainty; improvement of value accuracy; or change in distribution of encountered states. Some of these have the desirable property that if the proxy is zero, so is $LP$. However, these proxies themselves may only be available in approximated form, and these approximations tend to be highly dependent on the state distribution under which they are evaluated, which is subject to continual shift due to the changes in policy. As a result, direct comparison between different learning algorithms under these proxies tends to be precarious.

Second, our adaptive behaviour modulation can be viewed as an alternative to per-task hyper-parameter tuning, or hyper-parameter tuning with cross-task transfer (Golovin et al., 2017), and can be compared to other works attempting to reduce the need for this common practice. (Note that the best-fixed-arm in our experiments is equivalent to explicitly tuning the modulations as hyper-parameters.) Though often performed manually, hyper-parameter tuning can be improved by random search (Bergstra et al., 2011), but in either case requires many full training cycles, whereas our work optimises the modulations on-the-fly during a single training run.

Like our method, Population Based Training (PBT, Jaderberg et al., 2017) and meta-gradient RL (Andrychowicz et al., 2016; Xu et al., 2018) share the property of dynamically adapting hyper-parameters throughout agent training. However, these methods exist in a distinctly different problem setting: PBT assumes the ability to run multiple independent learners in parallel with separate experience. Its cost grows linearly with the population size (typically $> 10$), but it can tune other hyper-parameters than our approach (such as learning rates). Meta-gradient RL, on the other hand,

---

[1]Since it is too expensive to investigate all individual combinations in the joint modulation space, we only vary $z$-s along a single dimension at a time.

assumes that the fitness is a differentiable function of the hyper-parameters, which may not generally hold for exploration hyper-parameters.

While our method focuses on modulating behaviour in order to shape the experience stream for effective learning, a related but complementary approach is to filter or prioritize the generated experience when sampling from replay. Classically, replay prioritization has been based on TD error, a simple proxy for the learning progress conferred by an experience sample (Schaul et al., 2015). More recently, however, learned and thereby more adaptive prioritization schemes have been proposed (Zha et al., 2019), with (approximate) learning progress as the objective function.

# 7 DISCUSSION & FUTURE WORK

Reiterating one of our key observations: the qualitative properties of experience generated by an agent impact its learning, in a way that depends on characteristics of the task, current learning parameters, and the design of the agent and its learning algorithm. We have demonstrated that by adaptively using simple, direct modulations of the way an agent generates experience, we can improve the efficiency of learning by adapting to the dynamics of the learning process and the specific requirements of the task. Our proposed method[2] has the potential to accelerate RL research by reducing the burden of hyper-parameter tuning or the requirement for hand-designed strategies, and does so without incurring the computational overhead of some of the alternatives.

The work presented in this paper represents a first stab at exploiting adaptive modulations to the dynamics of learning, and there are many natural ways of extending this work. For instance, such an approach need not be constrained to draw only from experiences generated by the agent; the agent can also leverage demonstrations provided by humans or by other agents. Having an adaptive system control the use of data relieves system designers of the need to curate such data to be of high quality – an adaptive system can learn to simply ignore data sources that are not useful (or which have outlived their usefulness), as our bandit has done in the case of choosing modulations to generate experiences with (e.g., Figures 17, 18, 19).

A potential limitation of our proposal is the assumption that a modulation remains fixed for the duration of an episode. This restriction could be lifted, and one can imagine scenarios in which the modulation used might depend on time or the underlying state. For example, an agent might generate more useful exploratory experiences by having low stochasticity in the initial part of an episode, but switching to have higher entropy once it reaches an unexplored region of state space.

There is also considerable scope to expand the set of modulations used. A particularly promising avenue might be to consider adding noise in parameter space, and controlling the variance (Fortunato et al., 2018; Plappert et al., 2018). In addition, previous works have shown that agents can learn diverse behaviours conditioned on a latent policy embedding (Eysenbach et al., 2018; Haarnoja et al., 2018), goal (Ghosh et al., 2018; Nair et al., 2018) or task specification (Borsa et al., 2019). A bandit could potentially be exposed to modulating the choices in abstract task space, which could be a powerful driver for more directed exploration.

ACKNOWLEDGEMENTS

(omitted for anonymity)

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

# A  ADAPTIVE BANDIT

In this section we will briefly revisit the adaptive bandit proposed in Section 4 and provide more of the details behind the adaptability of its horizon. For clarity, we start by restating the setting and key quantities: the reference fitness $m_t$, the active horizon $h_t$ and the observed fitness function $f_t : \mathcal{Z} \mapsto \mathbb{R}$, where $\mathcal{Z} = \{z_1, \ldots, z_K\}$ defines a modulation class.

The bandit samples a modulation $z \in \{z_1, \ldots, z_K\}$ according to the probability that this $z$ will result in higher than average fitness (within a recent length-$h$ window):

$$\mathbb{P}_t(z) \propto \mathbb{P}(f_t(z) \geq m_t), \quad \text{where } m_t \doteq \frac{1}{h} \sum_{t'=t-h}^{t-1} f_{t'}(z_{t'}).$$

**Adapting $z$-probabilities.** For simplicity, $\mathbb{P}_t(z)$ is inferred based on the empirical data within a recent time window of the same horizon $h$ that is used to compute $m_t$. Concretely, $\mathbb{P}_t(z) \doteq \mu_t(z)/\sum_{z'} \mu_t(z')$ with the preferences $\mu_t(z) \approx \mathbb{P}(f_t(z) \geq m_t)$ defined as

$$\mu_t(z) \doteq \frac{\frac{1}{2} + \sum_{t'=t-h}^{t-1} \mathbb{I}_{f_{t'}(z'_t) \geq m_t} \mathbb{I}_{z_{t'}=z}}{1 + n(z,h)},$$

where $n(z,h)$ is the number of times that $z$ was chosen in the corresponding time window. We encode a prior preference of $\frac{1}{2}$ in the absence of other evidence, as an additional (fictitious) sample.

**Adapting the horizon.** As motivated in Section 4, the discrete horizon size $h_t = h_{t-1}+1$ is adapted in order to improve regression accuracy $\mathcal{L}_t(h)$:

$$\mathcal{L}_t(h) \doteq \frac{1}{2} \left( f_t(z_t) - \bar{f}_{t,h}(z_t) \right)^2,$$

where $f(z_t)$ is the fitness of the modulation $z_t$ chosen at time $t$, and

$$\bar{f}_{t,h}(z) \doteq \frac{m_t + \sum_{t'=t-h}^{t-1} f_t(z_{t'}) \mathbb{I}_{z_{t'}=z}}{1 + n(z,h)}.$$

This objective is not differentiable w.r.t. $h$, so we perform a finite-difference step. As the horizon cannot grow beyond the amount of available data, the finite-difference is not symmetric around $h_t$. Concretely, at every step we evaluate two candidates: one according to the current horizon $h_t$, $\mathcal{L}_t(h_t)$, and one proposing a shrinkage in the effective horizon, $\mathcal{L}_t(h')$, where the new candidate horizon is given by:

$$h' \doteq \max(2K, (1-\eta)h_t).$$

Thus the new horizon proposes a shrinkage of up to $\eta = 2\%$ per step, but is never allowed to shrink beyond twice the number of arms $K$. Given a current sample of the fitness function $f_t(z_t)$, we probe which of these two candidates $h_t$ or $h'$ best explains it, by comparing $\mathcal{L}_t(h_t)$ and $\mathcal{L}_t(h')$. If the shorter horizon seems to explain the new data point better, we interpret this as a sign of non-stationarity in the process and propose a shrinkage proportional to the relative error prediction reduction $\frac{\mathcal{L}_t(h_t)-\mathcal{L}_t(h')}{\mathcal{L}_t(h_t)}$, namely:

$$h_{t+1} = \begin{cases} \max\left(2K, (1 - \eta \frac{\mathcal{L}_t(h_t)-\mathcal{L}_t(h')}{\mathcal{L}_t(h_t)})h_t\right) & \text{if } \mathcal{L}_t(h_t) > \mathcal{L}_t(h') \\ h_t + 1 & \text{otherwise.} \end{cases}$$

**Factored bandits**. In the case of factored sub-bandits, they each maintain their own independent horizon $h$; we have not investigated whether sharing it would be beneficial.

# B  LAVAWORLD EXPERIMENTS

In this section we will describe in further details the experiments behind Figure 2. These were conducted on LavaWorld, a small (96 states), deterministic four-rooms-style navigation domain, with deadly lava instead of walls. We chose this domain to illustrate what our proposed adaptive mechanism (Section 4) would do under somewhat idealised conditions where the learning is tabular

and we can compute the ground-truth $LP(z)$ and assess oracle performance. This investigation allows us to first see how well the bandit can deal with the kind of non-stationary arising from an RL learning process (entangled with exploration).

In this setting, we consider three modulation classes, $\epsilon, T, \boldsymbol{b}$, where $\boldsymbol{b}$ can boost the logits for any of the 4 available actions. The sets of modulations are $\epsilon, T \in \{0.01, 0.1, 1\}$ and $\boldsymbol{b}_i \in \{0, 0.1\}$ resulting in 31 unique modulated (stochastic) policies for each Q-function (see Figure 8). Q-functions are look-up tables of size $(96 \times 4)$ as this domain contains 96 unique states. The single start state is in the top-left corner, and the single rewarding state is in the top-left corner of the top-right room, and is also absorbing. We treat the discount $\gamma = 0.99$ as probability of continuation, and terminate episodes stochastically based on this, or when the agent hits lava.

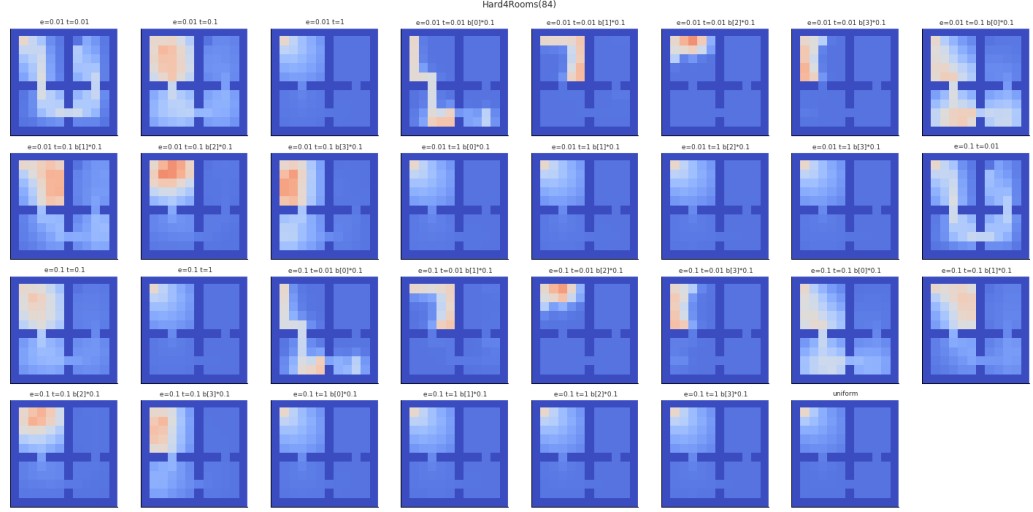

Figure 8: Example visitation densities in LavaWorld of different modulated policies that all use the same set of Q-values.

We study the behaviour of the system in two settings: one stationary, one non-stationary.

In the stationary setting (Figure 2, left), we considered modulation behaviours $\pi(\cdot|s, z)$ that do not change over time. They are computed based on a 'dummy' action-value function $Q$, as described in Section 2, but this value does not change over time. The learning process is tabular and independent of this behaviour generating value $Q$. In this case, we compute the cumulative probability that an executed policy encounters the single sparse reward ('expected reward') as a function of the number of episodes, where we assume that the policy will be perfect after the first reward event. The $LP(z)$ signal given to the oracle bandit is the true (stationary) expectation of this event for every modulation $z$. Given the extreme reward sparsity and the absence of learning, there is no obvious choice for a proxy measure $f$, so the non-oracle bandit reverts to a uniform choice over $z$. The reference Q-values are the optimal ones for the task. The results presented are averaged over 10 runs.

Secondly, we considered a non-stationary setting (Figure 2, right) similar to the one above, but where the Q-values behind the modulated behaviour are learned over time. This is akin to the actual regime of operation this system would encounter in practice, although the learning of these values is idealised. These are initialised at zero, and the only update (given the absence of rewards) is to suppress the Q-value of any encountered transition that hits lava by $0.1$; this learning update happens instantaneously. In other words, the agent does a random walk, but over time it dies in many different ways. The reported "expected reward" is again the probability that the policy induced by $Q_t$ encounters the final reward. In this case, a reasonable proxy $f(z)$ for $LP(z)$ is the binary signal on whether something was learned from the last episode (by encountering a new into-lava transition), which obtains a large fraction of the oracle's performance.

## C  ATARI HYPER-PARAMETERS

In this section we record the setting used in our Atari experiments: hyperparameters of the agents (Tables 2 and 3), environment and preprocessing (Table 1), and modulation sets for our behaviours (Table 4).

**Reference modulations**   Unless specified otherwise, we use the following modulations by default: $\epsilon = 0.01$, temperature $T = 0.00001$ (for tie-breaking between equal-valued actions), biases $\boldsymbol{b} = 0$, optimism $\omega = 0$, repeat probability $\rho = 0$. This corresponds to the most commonly used settings in the literature. On learning curve plots, this fixed setting is always shown in black.

**Modulation sets**   The sets of curated and non-curated modulations we use are described in Table 4. Curated values were chosen based on the results in Figure 4.

**Fixed hyperparameters**   The hyper-parameters used by our Atari agent are close to defaults used in the literature, with a few modification to improve learning stability in our highly distributed setting. For preprocessing and agent architecture, we use DQN settings detailed in Table 1 and Table 2. Table 3 summarizes the other hyper-parameters used by our Atari agent.

| Parameter | value |
|---|---|
| Grey-scaling | True |
| Observation down-sampling | (84, 84) |
| Frames stacked | 4 |
| Reward clipping | [-1, 1] |
| Action repeats (when $\rho = 0$) | 4 |
| Episode termination on loss of life | False |
| Max frames per episode | 108K |

Table 1: Environment and preprocessing hyperparameters.

| Parameter | value |
|---|---|
| channels | 32, 64, 64 |
| filter sizes | $8 \times 8, 4 \times 4, 3 \times 3$ |
| stride | 4, 2, 1 |
| hidden units | 512 |

Table 2: Q network hyperparameters.

## D  EVALUATION

In this section we detail the evaluation settings used for our Atari experiments, as well as the metrics used to aggregate results across games in Figure 5 in the main text.

**Games**   We evaluate on a set of 15 games chosen for their different learning characteristics: AS-TERIX, BREAKOUT, DEMON ATTACK, FROSTBITE, H.E.R.O., MS. PAC-MAN, PRIVATE EYE, Q*BERT, SEAQUEST, SPACE INVADERS, STAR GUNNER, TENNIS, VENTURE, YARS' REVENGE and ZAXXON.

Human-normalised scores are computed following the procedure in (Mnih et al., 2015), but differing in that we evaluate online (without interrupting training), average over policies with different weights $\theta_t$ (not freezing them), and aggregate over 20 million frames per point instead of 1 million.

**Metrics**   The relative rank statistic in Section 5.1 are normalized to fall between 0 and 1 for any set of outcomes $G(game, \cdot, variant)$ for any number of seeds $N$. For this we simply scale the raw average ranks by their minimal and maximal values: $\frac{N+1}{2}$ and $N^+ + \frac{N+1}{2}$ where $N^+$ is the number of other outcomes this variant is jointly ranked with.

| Parameter | value |
|---|---|
| $n$ for $n$-step learning | 3 |
| number of quantile values | 11 |
| prioritization type | proportional to absolute TD error |
| prioritization exponent $\alpha$ | 0.6 |
| prioritization importance sampling exponent $\beta$ | 0.3 |
| learning rate | 0.0001 |
| optimization algorithm | Adam |
| Adam $\epsilon$ setting | $10^{-6}$ |
| $L_2$ regularization coefficient | $10^{-6}$ |
| target network update period | 1K learner updates ($=$ 250K environment frames) |
| replay memory size | 1M transitions |
| min history to start learning | 80K frames |
| batch size | 512 |
| samples to insertion ratio | 8 |
| Huber loss parameter | 1 |
| max gradient norm | 10 |
| discount factor | 0.99 |
| number of actors | 40 |

Table 3: Agent hyperparameters.

| Modulation | Curated set | Extended set |
|---|---|---|
| Temperature $T$ | 0.0001, 0.001, 0.01 | 0.00001, 0.0001, 0.001, 0.01, 0.1, 1, 10 |
| $\epsilon$ | 0, 0.001, 0.01, 0.1 | 0, 0.001, 0.01, 0.1, 0.2, 0.5, 1 |
| Repeat probability $\rho$ | 0, 0.25, 0.5 | 0, 0.25, 0.5, 0.66, 0.75, 0.8, 0.9 |
| Optimism $\omega$ | -1, 0, 1, 2, 10 | -10, -2, -1, 0, 1, 2, 10 |
| Action biases $\boldsymbol{b}$ | $\boldsymbol{0}$ | -1, 0, 0.01, 0.1 |

Table 4: Modulation sets.

In Figure 5, we use a different metric to compare the effect of a chosen modulation early into the learning process. For this we compute $\mathbb{E}_s[G_z^{(s)}]$, the average episode return of modulation $z$ at the end of training across all seeds $s \in S$, and $z_0$, the modulation with highest average episode returns at the beginning of training (first 10% of the run). Based on this, we compute the normalised drop in performance resulting from committing prematurely to $z_0$:

$$\text{Performance drop}(z) \doteq \frac{\mathbb{E}_s[G_{z_0}^{(s)}] - \mathbb{E}_s[G_{z^-}^{(s)}]}{\mathbb{E}_s[G_{z^+}^{(s)}] - \mathbb{E}_s[G_{z^-}^{(s)}]}$$

where $z^+ = \arg\max_{z \in \mathcal{Z}} \mathbb{E}_s[G_z^{(s)}]$, $z^- = \arg\min_{z \in \mathcal{Z}} \mathbb{E}_s[G_z^{(s)}]$ and $\mathcal{Z}$ the modulation class considered in the study ($\epsilon$'s, temperatures $T$, action repeat probabilities $\rho$, and optimism $\omega$).

# E    ADDITIONAL ATARI RESULTS

## E.1    PER-TASK NON-STATIONARITY

In this section we report detailed results which were presented in aggregate in Figure 4. Specifically, these results show that (1) the most effective set of modulations varies by game, (2) different modulations are preferable on different games, and (3) the non-stationary bandit performs comparably with the best choice of modulation class on all games.

In the next few figures we give per-game performance comparing fixed-arm modulations with the adaptive bandit behaviour for modulation **epsilon** (Figure 9), **temperature** (Figure 10), **action repeats** (Figure 11), and **optimism** (Figure 12). For reference, we include the performance of a **uniform bandit** (dashed-line in the figures), over the same modulation set as the bandits, as well as the best parameter setting across games (**reference** solid black line in the figures).

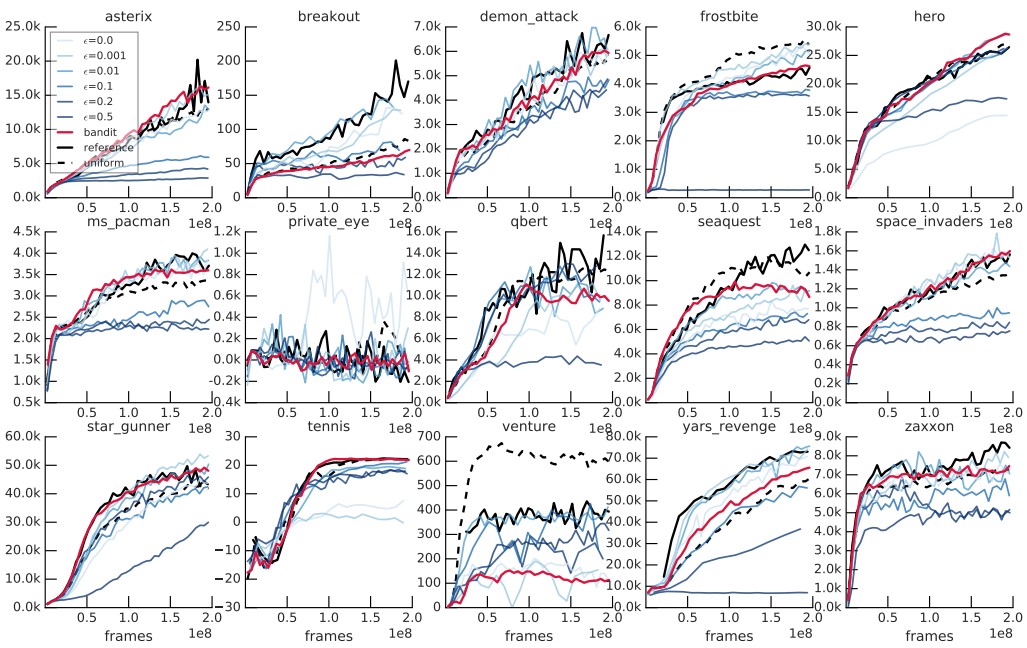

Figure 9: Fixed arms, bandit and uniform over the curated sets for **epsilon**.

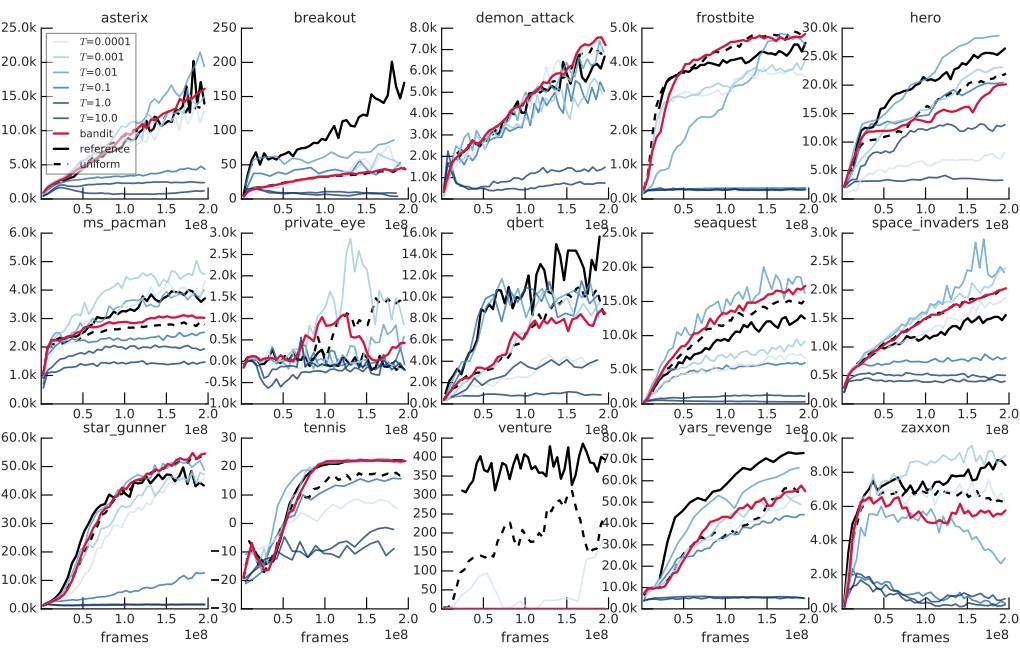

Figure 10: Fixed arms, bandit and uniform over the curated sets for **temperatures**.

## E.2 COMBINATORIAL BANDITS

In this section we include additional results for the different combinatorial bandits run on the curated and extended sets (see Table 4). Most of these experiments were run on subsets of the curated/extended sets across modulations, rather than the full Cartesian product. As a convention, whenever a modulation class is omitted from the experiment name, the value for this class is set to the default reference value reported in Section C (**Reference modulations**). Thus, for instance if we refer to a per-class-modulation bandit, say optimism $\omega$, the modulations $z$ for this class would

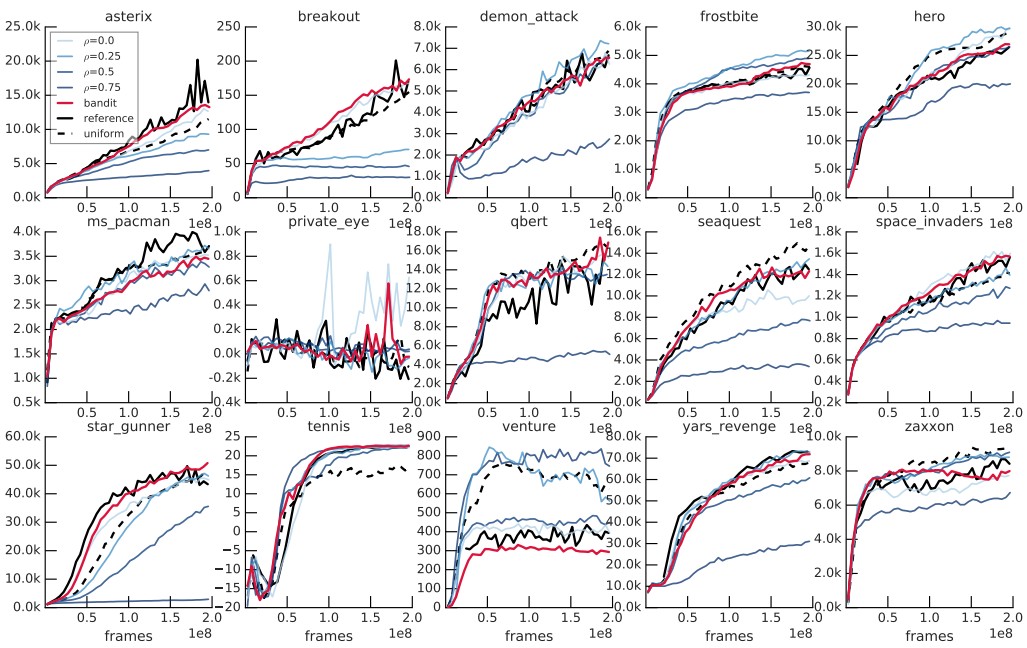

Figure 11: Fixed arms, bandit and uniform over the curated sets for **action repeats**.

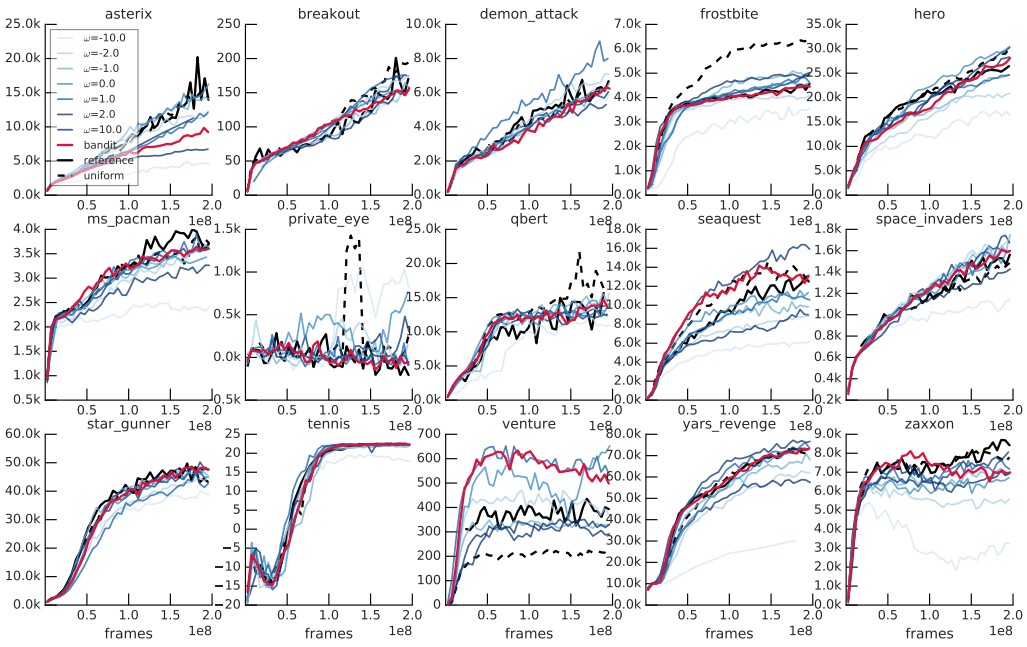

Figure 12: Fixed arms, bandit and uniform over the curated sets for **optimism**.

be the ones reported in Table 4 (line 4), while all other modulation dimensions would be kept fixed to their reference values.

Figure 13 shows that the combined bandit performs competitively compared with per-factor bandits (the same adaptive bandit but restricted to one class of modulation). In particular, it is worth noting that the per-factor bandit that performs best is game dependent. Nevertheless, the combined bandit, considering modulations across many of these dimensions, manages to recover a competitive performance across most games.

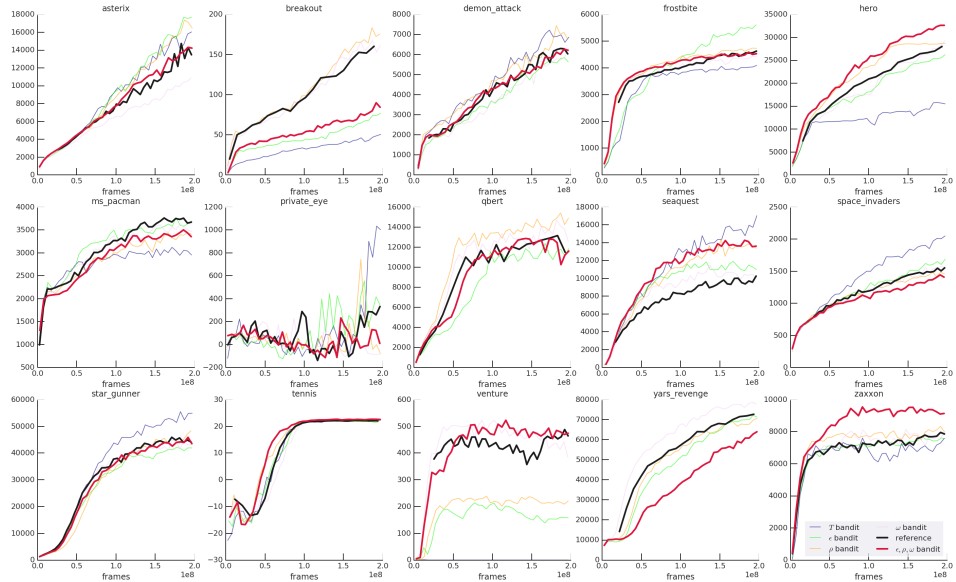

Figure 13: Comparing the combinatorial $\epsilon, \rho, \omega$-bandit to the per-modulation-class bandits.

In Figure 14, we include a comparison plot between the combinatorial bandit on the curated set of 3 modulation classes $(\epsilon, \rho, \omega)$, its uniform counterpart on the same set and the reference fixed arm across games. First thing to notice is that on the curated set the uniform bandit is quite competitive, validating our initial observation that the problem of tuning can be shifted a level above, by carefully curating a set of good candidates. We can additionally see that the adaptive mechanism tends to fall in-between these two extremes: an uninformed arm selection and tuned arm selection. We can see that the adaptive mechanism can recover a behaviour close to uniform in some games (H.E.R.O., YARS' REVENGE), while maintaining the ability to recover something akin to best arm identification in other games (see ASTERIX). Moreover there are (rare) instances, see ZAXXON, where the bandit outperforms both of these extremes.

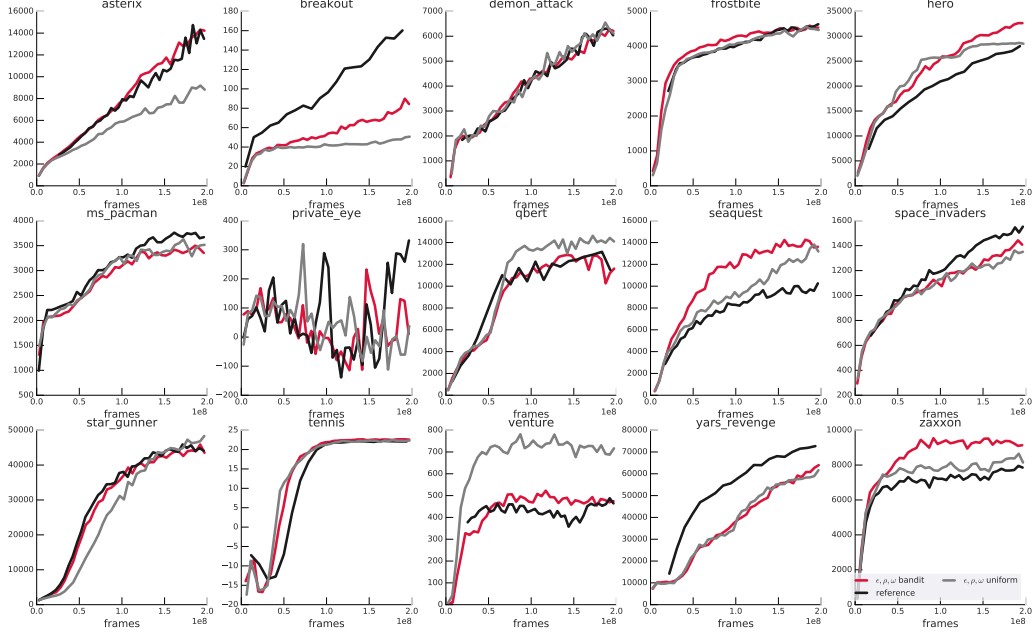

Figure 14: The learning curves for the uniform and the $(\epsilon, \rho, \omega)$-bandit on the curated set.

In Figure 15 we include a plot comparing the performance of a combinatorial bandit on the full curated and extended modulation sets. These are bandits acting on across all modulation classes outlined in Table 4. As a reference, we include the performance of the per-class modulation bandits, as in Figure 13. The bias modulation class was omitted as modulating exclusively within this class leads to very poor performance as policies tend to lock into a particular preference. We can also see a negative impact on the overall performance when adding a bias set to the set of modulations the bandit operates on, as one can see from Figure 15 (magenta line). This is why we opted not to include this set in the extended bandit experiments reported in Figure 6 and restricted ourselves to the other 4 extended modulation sets.

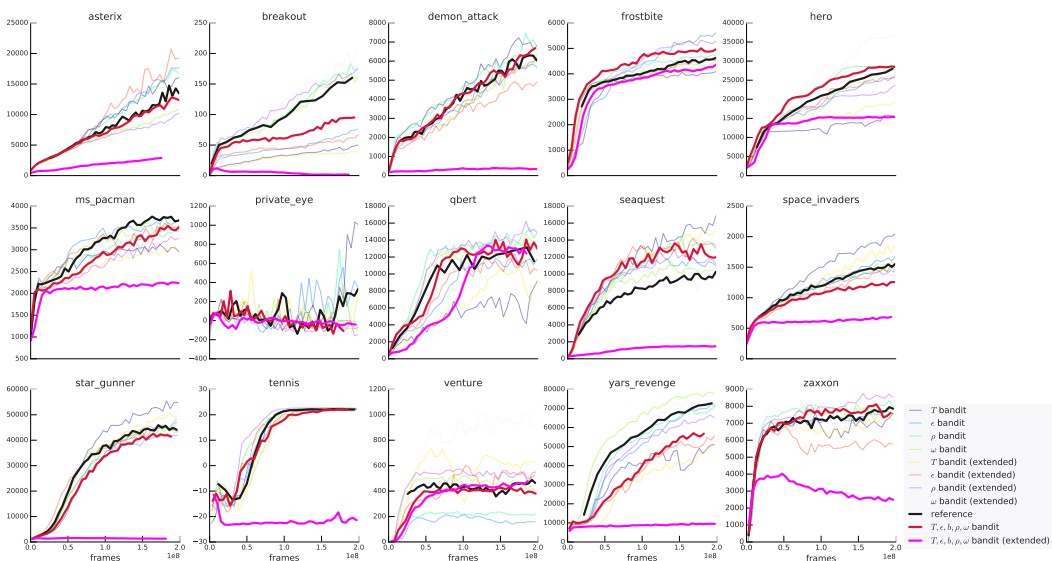

Figure 15: Bandits on extended sets versus curated sets. For single-class modulation, the gaps are small overall (as shown in Figure 6). One negative result is that on the fully combinatorial space $T, \epsilon, \boldsymbol{b}, \rho, \omega$, the bandit performance drops substantially (compare red to magenta). We assume this is due to the dominant effect of strong action biases $\boldsymbol{b}$ that effectively impoverish the behaviour for many sampled $z$.

### E.3    OTHER BANDIT BASELINES

Finally, in Figure 16 we provide a comparison to other more established bandit algorithms, UCB (Auer, 2002; Kaufmann et al., 2012) and Thompson Sampling (Thompson, 1933; Chapelle & Li, 2011), that would need to learn with modulation to use. The results in this figure are averaged across 5 seeds, and jointly modulate across three classes ($\epsilon, \rho, \omega$). We also include resulting learning curves for our proposed adaptation mechanism, the bandit described in Section 4, as well as uniform. The first thing to notice is that the stationary bandits, UCB and Thompson Sampling, are sometimes significantly worse than uniform, indicating that they prematurely lock into using modulations that may be good initially, but don't help for long-term performance. We have already seen signs of this non-stationarity in the analysis in Figure 5 which shows that early commitment based on the evidence seen in the first part of training can be premature and might hinder the overall performance. In contrast, our proposed bandit can adapt to the non-stationarity present in the learning process, resulting in performance that is on par or better than these baselines in most of these games (with only one exception, in YARS' REVENGE, where it matches the performance of the uniform bandit). In that context, it is worth highlighting that the *best* alternative baseline (UCB, Thompson Sampling, Uniform) differs from game to game, so outperforming all of them is a significant result. Another point is that UCB and Thompson Sampling still require some hyper-parameter tuning (we report the results for the best setting we found), and thus add extra tuning complexity, while our approach is hyper-parameter-free.

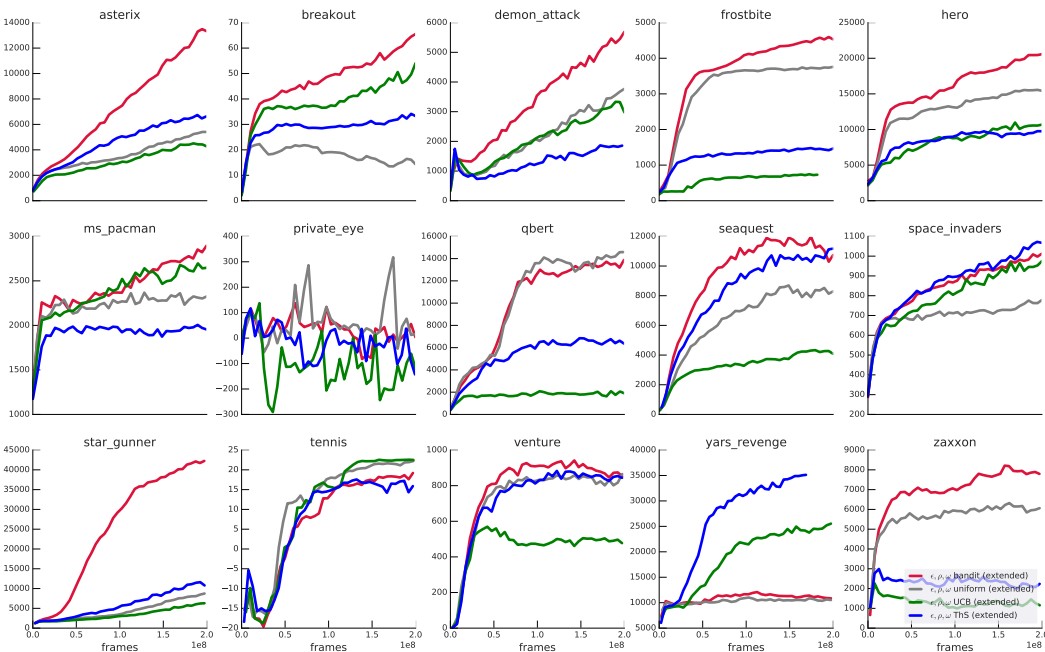

Figure 16: Comparison of several bandits (and uniform) on the combinatorial space spanned by the extended set of 3 modulations $\epsilon, \rho, \omega$. The performance of uniform (gray), UCB (green) and Thompson Sampling (blue) varies considerably from one game to another, while our proposed adaptive mechanism ('bandit', red) seems to deal quite effectively with this variability and achieves very good performance across almost all games. Note that the bandit and uniform results here correspond to the extended 'combo' results in Figure 4.

### E.4  BEHAVIOUR OF THE NON-STATIONARY BANDIT

In the previous section we saw that the bandit is able to effectively modulate the behaviour policy to give robust performance across games. In this section we dive in deeper to analyse the precise modulations applied by the bandit over time and per-game. We see that the bandit does indeed adaptively change the modulation over time and in a game-specific way.

With these results we aim to look into the choices of the bandit across the learning process, and in multiple games. Most of the effect seems to be present in early stages of training.

- **Epsilon schedules:** Figure 17 shows the evolution of the value of $\epsilon$ by the bandit over time. We see that this gives rise to an adaptive type of epsilon schedule, where early in training (usually the first few million frames) large values are preferred, and as training progresses smaller values are preferred. This leads to a gradually increasingly greedy behaviour policy.

- **Action repeats decay:** Figure 18 shows the bandit modulated values for action repeats. We observe that as the agent progresses it can benefit from more resolution in the policy. Thus, the agent adaptively moves to increasingly prefer low action repeat probability over time, with a prolonged period of non-preference early in training.

- **SEAQUEST:** Figure 19 shows the evolution of the sampling distributions of a combined bandit with access to all modulation dimensions: $T$, $\epsilon$, $\boldsymbol{b}$, $\rho$, and $\omega$. Despite having over 7.5 million combinations of modulation values, the bandit can efficiently learn the quality of different arms. For instance, the agent quickly learns to prefer the down action over the up action, to avoid extreme left/right biases, and to avoid suppressing the fire action, which is consistent with our intuition (in SEAQUEST, the player must move below the sea level to receive points, avoid the left and right boundaries, and fire at incoming enemies). Moreover, as in the case of single arm bandits discussed above, the combined bandit prefers more stochastic choices of $\epsilon$ and temperature in the beginning of training and more deterministic settings later in training, and the action repeat probability decays over time.

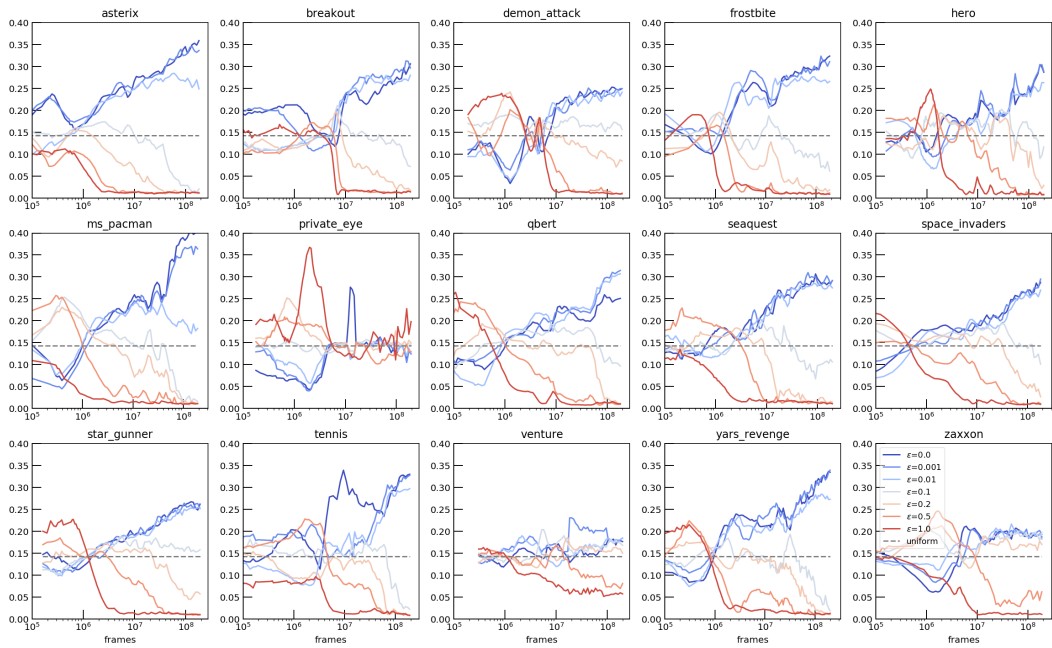

Figure 17: Choices of $\epsilon$ by the bandit across time (log-scale) on the extended set of $\epsilon$. Note that in general the values of $\epsilon$ start high and gradually diminish, but the emerging schedules are quite different across games.

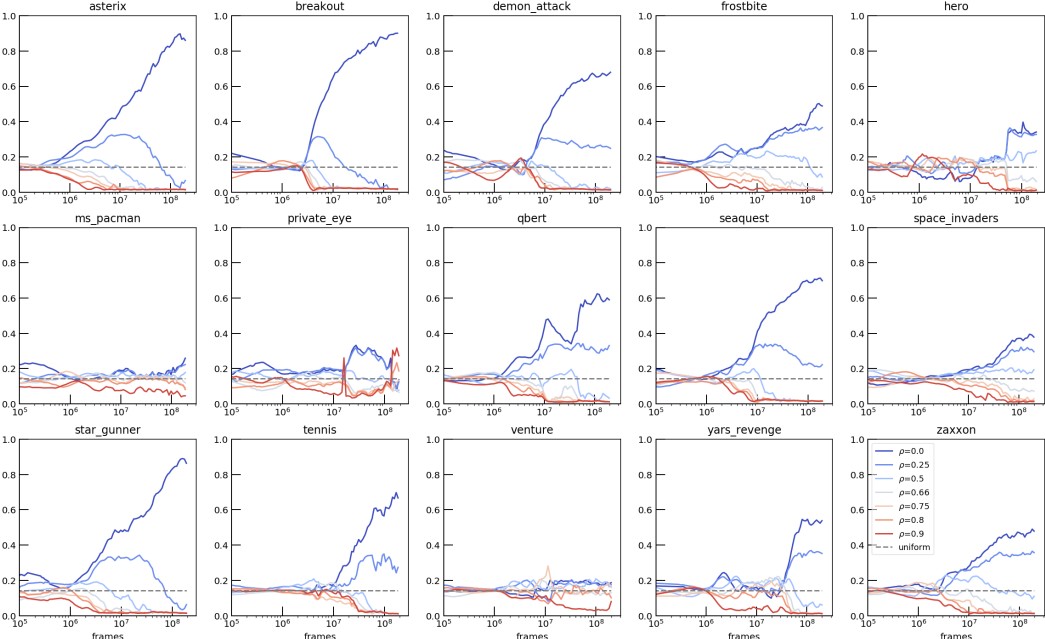

Figure 18: Choices of action repeat over time for different games on the extended set of $\rho$.

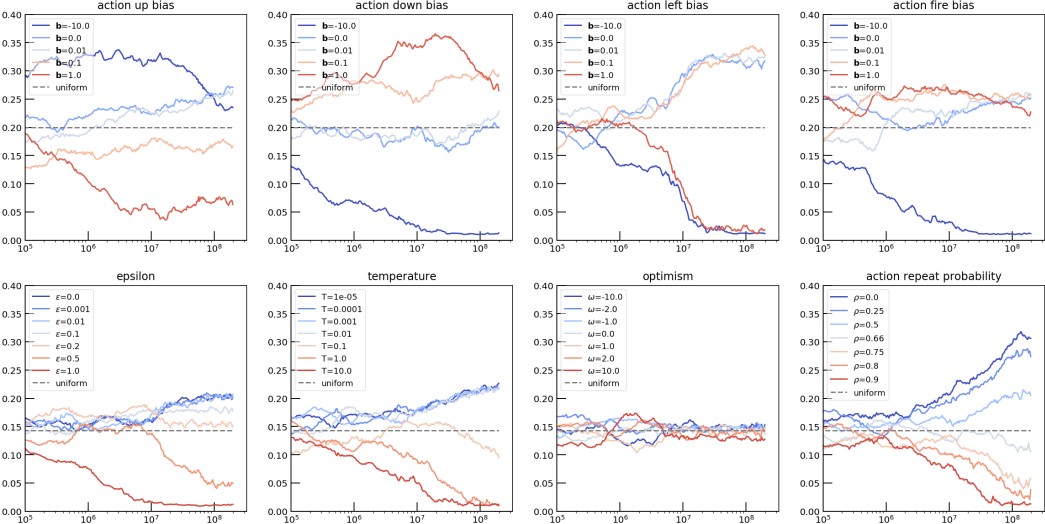

Figure 19: Probabilities of selected action bias scales $b$ (first four subplots) and other modulation dimensions across training, on the game of SEAQUEST.

