# OpenReview forum: "Adapting Behaviour for Learning Progress"
_ICLR.cc/2020/Conference — Reject_

### Official Review · AnonReviewer3 · 2019-10-23
**Official Blind Review #3**

**Rating:** 6

**Review:**

This paper presents an adaptive exploration scheme that can reduce the complexity of per-task tuning. This goal is achieve by formulating the adapting scheme as a multi-arm bandit problem with the actual "learning progress" as a feedback signal.

The paper is well written and easy to be understood.

The strength of this paper is that 1) the proposed method is new in the sense that it invents an automatic way for exploration. 2) The algorithm is simple yet effective by the experiment results the authors provide.

Weakness: the presentation of the tables/bar charts in the experiment is a bit unclear. More explanations are needed.

**Experience Assessment:**

I have published one or two papers in this area.

**Review Assessment: Checking Correctness Of Derivations And Theory:**

I assessed the sensibility of the derivations and theory.

**Review Assessment: Checking Correctness Of Experiments:**

I assessed the sensibility of the experiments.

**Review Assessment: Thoroughness In Paper Reading:**

I read the paper at least twice and used my best judgement in assessing the paper.

---

> ### Author Response · Authors · 2019-11-12
> **Response to reviewer #3**
>
> Thank you for your constructive feedback!
>
>
> > Weakness: the presentation of the tables/bar charts in the experiment is a bit unclear. More explanations are needed.
>
> Thank you for this suggestion (shared by other reviewers too): the updated version of the paper will clarify these things, and relegate less of the information to the appendix too. Specifically for Figure 4, the performance outcome for each variant is measured on multiple independent runs (seeds). All the outcomes are then jointly ranked, and the ranks are averaged across seeds. Finally, these averaged ranks are normalized to fall between 0 and 1. A normalized rank of 1 corresponds to all the N outcomes (seeds) of a variant being ranked at the top N positions in the joint ranking. Figure 4 then further aggregates these normalized ranks across 15 Atari games. Note that these joining rankings are done separately per subplot (ie modulation class).
>
>
> We think we could address all your concerns, but please let us know if you have further questions, the discussion period lasts until the end of the week!

---

> > ### Comment · AnonReviewer3 · 2019-11-14
> > **Response**
> >
> > Thank you for the response!  I will keep my rating after reading your response.

---

### Official Review · AnonReviewer2 · 2019-10-27
**Official Blind Review #2**

**Rating:** 3

**Review:**

This papers studies how to explore, in order to generate experience for faster learning of policies in context of RL. RL methods typically employ simple hand-tuned exploration schedules (such as epsilon greedy exploration, and changing the epsilon as training proceeds). This paper proposes a scheme for learning this schedule. The paper does this by modeling this as a non-stationary multi-arm bandit problem. Different exploration settings (tuple of choice of exploration, and the exact hyper-parameter), are considered as different non-stationary multi-arm bandits (while also employing some factorization) and expected returns are maintained over training. Arm (exploration strategy and hyper-parameter) is picked according to the return. The paper demonstrates results on the Atari suite of RL benchmarks, and shows results that demonstrate that their proposed search leads to faster learning.

Strength:
1. The paper tackles an interesting and important problem. The proposed solution is simple, yet effective.

Shortcomings:
1. The presentation is somewhat convoluted. The paper motivates the problem that we need to pick out an exploration sequence that optimizes learning progress, but then approximates it as simply measuring the return. Given there is no theoretical justification for the approximation, I believe the paper claims more than what it delivers and should change the presentation, so as not to claim that it is measuring and capturing learning progress to learn faster.

2. I am confused by Figure 4, and in general with the relative rank metrics. Specifically, in Figure 4, is it that the proposed bandit approach not as good as picking a single hyper-parameter for the different settings (T=0.01, eps=0.01, omega=2.0)? Similarly, for Figure 2, a singe fixed z, seems to do better than the bandit versions. Why doesn't the proposed bandit algorithm not pick out the best hyper-parameter? How well would a simpler hyper-parameter search procedure (picking the best hyper-parameter after the first 2000 episodes)?

3. This apart, I think that the experiment section is pretty hard to read, given all the metrics and methodology is in the Appendix. An alternate organization that presents all the main results in the main body in a self-contained manner will help.

4. Comparison with past works. I believe there are other existing works that should be cited and compared to. Using bandits to decide between different hyper-parameters is common (for example, see [A] for a service to do this with ML models), [B] uses improvements in accuracy as a way to pick between which question type to train on. Such past works should be cited and compared against.

[A] https://ai.google/research/pubs/pub46180
[B] Learning by Asking Questions
Ishan Misra, Ross Girshick, Rob Fergus, Martial Hebert, Abhinav Gupta and Laurens van der Maaten

**Experience Assessment:**

I have read many papers in this area.

**Review Assessment: Checking Correctness Of Derivations And Theory:**

N/A

**Review Assessment: Checking Correctness Of Experiments:**

I assessed the sensibility of the experiments.

**Review Assessment: Thoroughness In Paper Reading:**

I read the paper at least twice and used my best judgement in assessing the paper.

---

> ### Author Response · Authors · 2019-11-12
> **Response to reviewer #2**
>
> Thank you for your constructive feedback!
>
>
> Comment 1:
> We acknowledge that our presentation focused more than necessary on ideal scenarios that use learning progress LP(z) while the practical version used a (maybe disappointingly) simplistic choice of proxy f(z). The updated paper will change the emphasis, and clarify that a proper learning progress proxy remains future work.
> We will also clarify that the little phrase “After initial experimentation, we opted for the simple proxy…” implies quite extensive experimentation with other plausible proxies that looked promising in individual environments but were not consistently effective across the suite of Atari games.
>
>
> Comment 2:
> Sorry, our presentation of Figure 4 was not very clear: The performance outcome for each variant is measured on multiple independent runs (seeds). All the outcomes are then jointly ranked, and the ranks are averaged across seeds. Finally, these averaged ranks are normalized to fall between 0 and 1. A normalized rank of 1 corresponds to all the N outcomes (seeds) of a variant being ranked at the top N positions in the joint ranking. Figure 4 then further aggregates these normalized ranks across 15 Atari games. Note that these joining rankings are done separately per subplot (ie modulation class).
> The bandit is not guaranteed to reproduce the performance of the best arm for a couple of reasons: (a) the signal f(z) it obtains is noisy, (b) if is myopic in that it reflects only current performance not future learning, and (c) the dynamics are non-stationary, so the best arm changes over time. For all these reasons, the bandit we use is a conservative one that tends to spread the probability mass among decent-looking arms, while suppressing obviously sub-optimal arms.
> The experiment you suggest (picking the best hyper-parameter after the first X episodes) is exactly what we investigated in Figure 5 (left subplot). The empirical result is that it works well for some games but not others, and better for some modulation classes than others, but overall it’s not reliable. The updated paper will split Figure 5 into two to increase clarity.
>
>
> Comment 3:
> Thank you for that suggestion: we will update the organization of the paper to make the main body more self-contained.
>
>
> Comment 4:
> The updated paper will discuss related work in more depth, including the suggested [A] and [B].
>
>
> We think we could address all your concerns, but please let us know if you have further questions, the discussion period lasts until the end of the week!

---

### Official Review · AnonReviewer1 · 2019-10-30
**Official Blind Review #1**

**Rating:** 3

**Review:**

This paper develops a multi-arm bandit-based algorithm to dynamically adapt the exploration policy for reinforcement learning. The arms of the bandit are parameters of the policy such as exploration noise, per-action biases etc. A proxy fitness metric is defined that measures the return of the trajectories upon perturbations of the policy z; the bandit then samples perturbations z that are better than the average fitness of the past few perturbations.

I think this paper is just below the acceptance threshold. My reservations and comments are as follows.

1. While I see the value in designing an automatic exploration mechanism, the complexity of the underlying approach makes the contribution of the bandit-based algorithm difficult to discern from the large number of other bells and whistles in the experiments. For instance, the authors use Rainbow as  the base algorithm upon which they add on the exploration. Rainbow itself is an extremely complicated algorithm, how can one be certain that the improvements in performance are caused by the improved exploration and not a combination of the bandit’s actions with the specifics of Rainbow?
2. I don’t understand Figure 4. The score defined in Appendix is the average over games for which seed performs better. Why is the random seed being used to compare the performance of different arms? Do you instead mean that s and s’ are two values of the arm in Figure 4? If not, how should one interpret Figure 4, no fixed arm is always good because the performance varies across the seeds. The curated bandit does not seem to be doing any better than a fixed arm.

I have a few more questions that I would like the authors to address in their rebuttal or the paper.

1. The proxy f(z) does not bear any resemblance to LP(z). Why discuss the LP(z) then. The way f(z) is defined, it is just the value function averaged over perturbations  of the policy. If one were to consider z as an additional action space that is available to the agent during exploration, f(z) is the value function itself. The exploration policy is chosen not to maximize the E_z [f(z)] directly but to maximize the lower bound in Markov’s inequality (P(f(z) >= t) <= E_z [f(z)]/t) in Section 4.
2. Can you elaborate more on the metric for measuring the learning progress LP? Why does the myopic metric make sense in spite of the there being plateaus in the training curves?
3. The key contribution of the paper that the authors could highlight better is that they do not add new hyper-parameters. In this aspect, the auto-tuner for exploration is a plug-and-play procedure in other RL algorithms.
4. From Figure 6 and Figure 8-11, it looks like the bandit is more or less on par with fixed exploration policies. What is the benefit of the added complexity?

**Experience Assessment:**

I have published one or two papers in this area.

**Review Assessment: Checking Correctness Of Derivations And Theory:**

I carefully checked the derivations and theory.

**Review Assessment: Checking Correctness Of Experiments:**

I carefully checked the experiments.

**Review Assessment: Thoroughness In Paper Reading:**

I read the paper thoroughly.

---

> ### Author Response · Authors · 2019-11-12
> **Response to reviewer #1**
>
> Thank you for your constructive feedback!
>
>
> Main comment 1:
> Absolutely, this is a difficult issue: there is no perfect middle ground where it is possible to study the contributions in their simplest instantiations while at the same time verifying their practical effectiveness. We have opted to place the bulk of our emphasis on a realistic scenario (Atari with a Rainbow-like learning agent) that practitioners of Deep RL would find relevant. To isolate effects, our experimental section includes many variants and ablations, allowing us to state with confidence that modulating behaviour using the bandit improves performance compared to uniform (no bandit) or untuned (fixed modulation) baselines. And this is separately validated across multiple classes of modulations. But indeed, as you point out, we cannot guarantee that the improvements we see are purely due to exploration. At the same time, it’s worth recognising that, by design, the method proposed will try to cater to the underlying learning algorithm and would ideally generate samples that would benefit the underlying learning procedure. We will highlight this ambiguity in the revised paper.
>
>
> Main comment 2:
> Sorry, this was not very clear: The performance outcome for each variant is measured on multiple independent runs (seeds). All outcomes are then jointly ranked, and the corresponding ranks are averaged across seeds. Finally, these averaged ranks are normalized to fall between 0 and 1. A normalized rank of 1 corresponds to all the N outcomes (seeds) of a variant being ranked at the top N positions in the joint ranking. Figure 4 then further aggregates these normalized ranks across 15 Atari games. Note that these joining rankings are done separately per subplot (ie modulation class).
> Thus the reason that no fixed arm is always good does not depend on the inter-seed variability as much as on the fact that the best arm differs in different games. We will clarify this in the caption too.
> The bandit does not generally do better than the best fixed arm in hindsight -- in general, this would still need to be identified --  but it is not far off, and it handily outperforms untuned arms, allowing us to remove some of the hyper-parameter tuning burden.
>
>
> Additional question 1:
> We acknowledge that our presentation focused maybe more than necessary on ideal scenarios that use learning progress LP(z) while the practical version used a (maybe disappointingly) simplistic choice of proxy f(z). The updated paper will change the emphasis, and clarify that a closer, more faithful, learning progress proxy remains future work. We will also clarify that the little phrase “After initial experimentation, we opted for the simple proxy…” implies quite extensive experimentation with other plausible proxies that looked promising in individual environments but were not consistently effective across the suite of Atari games.
>
>
> Additional question 2:
> Of course, even an ideal metric LP(z) would remain a local quantity, and pursuing it would not guarantee the maximal final performance -- but it is valuable if local optima are not the prime concern.
> Performance plateaus are a nuisance in general, and within the simple space of modulations we consider, there is no magic bullet to escape them. However, our approach does the next best thing: when performance becomes an uninformative (ie on a plateau), it encourages maximal diversity of behaviour (tending toward uniform probabilities over z), with the hope that some modulation gets lucky -- and then as soon as that happens, very quickly focusing on that modulation to repeat the lucky episode until learning is progressing again.
>
>
> Additional question 3:
> Indeed, thank you. We have updated the text to place more emphasis on this contribution.
>
>
> Additional question 4:
> The way we would summarize these results is that the bandit is more or less on par with the *best* fixed exploration policy, and so the added complexity is justified by reducing the need to tune exploration. Is this what you meant?
>
>
> We think we could address all your concerns, but please let us know if you have further questions, the discussion period lasts until the end of the week!

---

### Official Review · AnonReviewer4 · 2019-11-01
**Official Blind Review #4**

**Rating:** 3

**Review:**

This in an interesting paper as it tries to alleviate the burden of hyper-parameters tuning for exploration strategies Deep Reinforcement learning.
The paper proposes an adaptive behaviour in order to shape the data generation process for effective learning. The paper considers a behaviour policy that is parametrized by a set of variables z called modulations: for example the Boltzmann softmax temperature, the probability epsilon for epsilon-greedy, per-action biases, ..
The author frame the modulations search into a non-stationary multi-armed bandit problem and proposes to adapt the modulations according to a proxy to the learning progress. The author provides thorough experimental results.

Comments:

- All the variations considered for the behaviour policy performs only myopic exploration and thus provably inefficient in RL.
- The proposed proxy is simply the empirical episodic return. It is not well explained in the paper how this proxy correlates with the Learning progress criteria.
- The proxy seems to encourage selecting modulations that lead to generate most rewarding trajectories. How this proxy incentives the agent to explore poorly-understood regions? In other terms, how this proxy help to tradeoff between exploration and exploitation ?
-  The modulation adaptation problem is framed into non-stationary multi-armed bandit problem but the authors present a heuristic to solve it instead of using provably efficient bandit algorithm such as exponential weight methods (Besbes et al 2014) or Thompson sampling (Raj & Kalyani 2017) cited in the paper.
- The way the authors adapt the modulation z (or at least its description in the paper) seems not technically sounded for me. They estimate a certain probability at time step t by empirical frequency based on data from previous time steps. But as the parameters change during the learning, the f_t’(z) at time t’ < t is not distributed as f_t(z). This introduces a biases in the estimate.
- I appreciate the thorough empirical results and ablation studies in the main paper and the appendix. They are really interesting.
- I am confused what is the fixed reference in Figure 6. It is not explained in the main paper. Is it a baseline with the best hyperprameters in hindsight?
-  From the plots of learning curves in appendix, the proposed methods doesn’t seem to show a huge boost of performance comparing to the uniform bandit. Could you show aggregated comparison between the proposed method and uniform bandit similarly to what is done in Figure 4 ?

**Experience Assessment:**

I have read many papers in this area.

**Review Assessment: Checking Correctness Of Derivations And Theory:**

N/A

**Review Assessment: Checking Correctness Of Experiments:**

I assessed the sensibility of the experiments.

**Review Assessment: Thoroughness In Paper Reading:**

I read the paper at least twice and used my best judgement in assessing the paper.

---

> ### Author Response · Authors · 2019-11-15
> **Response to reviewer #4**
>
> > All the variations considered for the behaviour policy performs only myopic exploration and thus provably inefficient in RL.
>
> Yes, we experiment only with myopic variants of exploration, but (A) our approach is not limited to this initial set of behaviour modulations, and could be extended to trade off between intrinsic and extrinsic motivation, or between model-free and model-based mechanisms; and (B) the variations we consider may not be ideal, but they are the ones most commonly used in domains like Atari.
>
>
> > The proposed proxy is simply the empirical episodic return. It is not well explained in the paper how this proxy correlates with the Learning progress criteria.  The proxy seems to encourage selecting modulations that lead to generate most rewarding trajectories. How this proxy incentives the agent to explore poorly-understood regions? In other terms, how this proxy help to tradeoff between exploration and exploitation ?
>
> Thank you for this suggestion, we have now clarified this connection in Section 3. We acknowledge that f departs from LP in a number of ways.
> First, it does not contain learner-subjective information, but this is partly mitigated through the joint use of with prioritised replay that over-samples high error experience. Another potential mechanism by which the episodic return can be indicative of future learning is because an improved policy tends to be preceded by some higher-return episodes -- in general, there is a lag between best-seen performance and reliably reproducing it.
> Second, the fitness is based on absolute returns not differences in returns as suggested by Equation 1; this makes no difference to the relative orderings of z (and the resulting probabilities induced by the bandit), but it has the benefit that the non-stationarity takes a different form: a difference-based metric will appear stationary if the policy performance keeps increasing at a steady rate, but such a policy must be changing significantly to achieve that progress, and therefore the selection mechanism should keep revisiting other modulations. In contrast, our absolute fitness naturally has this effect when paired with a non-stationary bandit.
> We have also updated the paper to highlight that our proposed proxy is to be understood as an initial, simple, working instance, with a lot of remaining future work that could extend and refine it.
>
>
> >  The modulation adaptation problem is framed into non-stationary multi-armed bandit problem but the authors present a heuristic to solve it instead of using provably efficient bandit algorithm such as [...]
>
> Thank you for the suggestion! We had experimented with a few of these variants before designing the proposed adaptation method. We have now included such a plot in the paper, comparing our method to UBC and Thompson sampling (Appendix E.3 and Figure 16). As you can see from this comparison, the performance of these well-known bandits depends on the game, and it is subject to tuning, which is what we wanted to avoid in the first place. In most games our bandit performs significantly better.
>
>
> > The way the authors adapt the modulation z (or at least its description in the paper) seems not technically sounded for me  [...]
>
> The distribution of f(z) does change as a function of the parameter change and thus as a function of time. This is precisely the kind of non-stationarity that our adaptive mechanism has to deal with. This is also the reason behind the adaptive window used in this work. In a sense, one can see the size of the window as a proxy for the effective time horizon at which things can be seen as stationary in the learning. The window over which we integrate evidence is chosen to make the best recommendation; thus every time we deviate too much from the sample distribution captured within it, we consider this as a sign of non-stationarity and shrink the window. This is by no means optimal, nor do we claim it is, but it seems to be a reliable enough proxy to outperform candidates that do assume stationarity (as portrayed by the comparison in Figure 16).
>
>
> > I am confused what is the fixed reference in Figure 6. It is not explained in the main paper. Is it a baseline with the best hyperprameters in hindsight?
>
> The “fixed reference” is described in Appendix C, and corresponds to the most commonly used settings in the literature. We made this clear in the main body of the text.
>
>
> >  From the plots of learning curves in appendix, the proposed methods doesn’t seem to show a huge boost of performance comparing to the uniform bandit. Could you show aggregated comparison between the proposed method and uniform bandit similarly to what is done in Figure 4 ?
>
> Yes, we show this in aggregate in Figure 6 (old Figure 5-right): it shows how the bandit is roughly on par with uniform when the modulation set is curated, but the bandit significantly outperforms uniform in the untuned (“extended”) setting. We clarified the caption for this too.

---

### Author Response · Authors · 2019-11-15
**Updated paper**

We would like to thank all four reviewers for the many constructive comments.

We uploaded a revised version of the paper now, and invite you to give it a second look, to see whether our changes, together with the individual reviewer responses have addressed your concerns.

Summary of changes:
- Additional results, comparing to standard bandits (UCB and Thompson sampling) that have lower performance, as expected, because they do not take non-stationarity into account. See the new Figure 16 and Appendix E.3.
- Clarified the aggregate “relative rank” metric that is used in Figure 4 and 5 in the main text, and clarified captions.
- Split Figure 5 into two separate figures and expanded captions.
- Described the fixed reference (solid black learning curves) in the main text.
- Additional suggested citations and related them to our work.
- More detailed explanation of the limitations and properties of the learning progress proxy that we use (section 3).
- Increased the emphasis on hyper-parameter reduction, and decreased the emphasis on learning progress.

---

### Decision · Program_Chairs · 2019-12-19

**Decision:**

Reject

**Comment:**

The paper introduces a non-stationary bandit strategy for adapting the exploration rate in Deep RL algorithms. They consider exploration algorithms with a tunable parameter (e.g. the epsilon probability in epsilon-greedy) and attempt to adjust this parameter in an online fashion using a proxy to the learning progress. The proposed approach is empirically compared with using fixed exploration parameters and adjusting the parameter using a bandit strategy that doesn't model the learning process.

Unfortunately, the proposed approach is not theoretically grounded and the experiments lack comparison with good baselines in order to be convincing. A comparison with other, provably efficient, non-stationary bandit algorithms such as exponential weight methods (Besbes et al 2014) or Thompson sampling (Raj & Kalyani 2017), which are cited in the paper, is missing. Moreover, given the whole set of results and how they are presented, the improvement due to the proposed method is not clear. In light of these concerns I recommend to reject this paper.